# Tibial Spine Height Measured by Radiograph Is a Risk Factor for Non-Contact Anterior Cruciate Ligament Injury in Males: A Retrospective Case-Control Study

**DOI:** 10.3390/ijerph192315589

**Published:** 2022-11-24

**Authors:** Shixin Nie, Jiaxing Chen, Hua Zhang, Pei Zhao, Wei Huang

**Affiliations:** 1Department of Orthopedics, The First Affiliated Hospital of Chongqing Medical University, Chongqing 400016, China; 2Orthopedic Laboratory of Chongqing Medical University, Chongqing 400016, China

**Keywords:** tibial spine height, tibial spine width, ACL injury, risk factor, male

## Abstract

Various anatomic abnormalities are implicated in non-contact anterior cruciate ligament (ACL) injury, but researchers rarely deal with the relation between tibial spine height and ACL injury. We conducted a retrospective case-control study to include 96 patients with and without non-contact ACL injuries. Tibial plateau width (TPW), medial and lateral tibial spine height (MTSH and LTSH), and tibial spine width (TSW) were measured by radiographs. The parameters were compared among subgroups. Binary regression mode, receiver operating characteristic curves, and the area under the curve (AUC) were used to evaluate the specific correlation of the parameters with ACL injury. As a result, we found that the ratio of LTSH/TPW was larger in ACL-injured patients than in ACL-intact controls (*p* = 0.015). In the study group, LTSH/TPW (*p* = 0.007) and MTSH/TPW (*p* = 0.002) were larger in males than in females. The ratio of LTSH/TPW had an AUC of 0.60 and a significant OR of 1.3 for ACL injury in males, but not in females. In conclusion, LTSH was larger in patients with ACL injury and is a risk factor for ACL injury in males. The impact of increased LTSH on the impingement between the grafts and lateral tibial spine during ACL reconstruction warrants further investigation.

## 1. Introduction

Anterior cruciate ligament (ACL) injury is a common sports-related disorder with an incidence of 30–78 per 100,000 people [1,2,3]. ACL injuries cost as much as a billion dollars each year in the United States and increase the financial outlay [4]. Various risk factors are implicated in ACL injury, such as biomechanical or anatomical features and hormonal factors [5,6].

Morphological abnormalities of the distal femur and proximal tibia are of great significance in contributing to non-contact ACL injury [7]. The lateral femoral condyle ratio and intercondylar notch width are significantly associated with ACL injury [8,9,10]. In addition, accumulating evidence has shown that an increased posterior tibial slope (PTS), shallow medial plateau depth, and decreased medial tibial spine volume are risk factors for ACL injury [11,12,13,14,15,16,17]. On the other hand, the PTS is responsible for ACL injury in females, rather than in males [18,19]. Differences in anatomic parameters between males and females should not be neglected.

In spite of the results previously reported, the literature regarding the relation between the morphology of the tibial spine and ACL injury is scarce. The inner part of the ACL tibial footprint is closely connected to the medial tibial spine [20,21]. Lansdown et al. [13] reported that the tibial spine is in close contact with the femoral intercondylar notch, which could change the tension acting on the ACL. Sturnick et al. and Iriuchishima et al. found that lateral tibial spine height (LTSH) was not correlated with ACL injury [15,22]. However, they did not consider the impact of individual differences (bone size) on the measurement of tibial spine height and the relatively small sample size is insufficient to specify the tibial spine morphology. On the other hand, impingement between the graft and lateral tibial spine during ACL reconstruction sometimes occurs [23], indicating that patients may be accompanied by an abnormal lateral tibial spine.

The hypothesis of this study was that increased LTSH was responsible for ACL injury and varied from males to females. Given the limitations of previous research, the purpose of this study was twofold: first, to verify the relationship between primary non-contact ACL injury and tibial spine height; second, to verify whether the relationships varied from males to females.

## 2. Materials and Methods

### 2.1. Study Population

This retrospective case-control study was approved by the institutional review board (IRB) of our hospital and the informed consent statement was waived (IRB NO. 2020-405). A total of 142 patients with ACL injuries and skeletal maturity who underwent surgical treatment from January 2017 to January 2019 in our institution were initially identified, and the radiographs were retrospectively collected. All the patients were verified as having ACL injury by magnetic resonance imaging (MRI) and arthroscopy surgery. The exclusion criteria were as follows: patients with contact ACL injury or with multiple ligament injuries (*n =* 6), patients with chronic ACL injury (*n =* 12) [24], patients with osteoarthritic changes in the knee (Kellgren-Lawrence III-IV, *n =* 4), patients who received ACL revision surgery (*n =* 2), patients with non-standard knee radiographs (with malrotation or flexion of the knee joints, *n =* 20), and patients with a history of previous knee trauma or surgery (*n =* 2). As a result, 96 patients were designated as the study group.

Then, the control group was built to include the subjects who referred to our institution for meniscus injury during the same period. Individuals aged 18 to 50 were randomly selected. The medical history of all the subjects in the control group was searched in the electronic medical record system, and the medical history and radiological data of all the participants were reviewed by an experienced orthopedist to ensure their eligibility for this study. MRI and arthroscopy surgery demonstrated that none of these patients had discoid meniscus or ACL injuries. As a result, 96 individuals with intact ACL were assigned as the control group.

### 2.2. Radiograph Technique

Images were obtained with a digital medical diagnostic X-ray fluoroscopy system (Shimadzu, Beijing, China). All the subjects underwent the radiograph examination in a standing position with the knee at full extension and close to the detector, with the patella facing forward, in order to eliminate the influence of malrotation and flexion of the knee joint on the measuring results. The X-ray bulb was taken at a caudal inclination of 10–15° to produce a complete intercondylar spine pattern with the anterior and posterior edges of the tibial plateau overlapped [25]. The scanning parameters were as follows: tube voltage, 60 kV; tube current, 250 mA, 16 ms; and a film distance of 120 cm.

### 2.3. Measurements

To assess the inter-observer reliability of the measurements, the morphological parameters of the tibial spine were evaluated by two well-trained orthopedic surgeons in a blinded and randomized fashion using the picture archiving and communication system (PACS). The anatomic parameters were measured by radiographs of the knee joints according to the method previously described [22], including the transverse width of tibial plateau (TPW), the height of the medial tibial spine (MTSH), the LTSH, and the horizontal distance between the top of the medial and lateral tibial spines (tibial spine width, TSW) (Figure 1). The ratios of TSW/TPW, LTSH/TPW, and MTSH/TPW were used to normalize these length parameters.

### 2.4. Statistical Analysis

The average values of the variables measured by both observers were input into SPSS software (version 26.0, IBM, Armonk, NY, USA) for statistical analysis. The intraclass correlation coefficient (ICC) was calculated to identify the inter-observer reliability of each measurement, with a value of >0.75 indicating excellent agreement. The Shapiro-Wilk normality test was performed to verify whether the data conformed to normal distribution. The independent sample *t*-test, Wilcoxon rank-sum test, and Chi-square test were conducted to compare data between groups. A binary logistic regression model (unadjusted) was established to identify the specific relationships between the parameters and ACL injury. To assess the diagnostic capacity of the parameters for ACL injury, receiver operating characteristic curves (ROC) and the area under the curve (AUC) were conducted via GraphPad Software (version 8.0.2, San Diego, CA, USA). The cut-off value of a parameter with an AUC greater than 0.70 was identified by Youden index. We set α at 0.05.

G-Power software (version 3.1.9, Heinrich-Heine-Universitat Dusseldorf, Dusseldorf, Germany) was used to calculate the power of this study. For the effect size of 1.07 according to the difference in the TSW between the ACL-injured patients and ACL-intact controls, a power of 1.00 was calculated (n, 96; alpha, 0.05).

## 3. Results

Included in this study were 96 patients with non-contact ACL injury (37 females and 59 males, mean age ± SD 29.5 ± 8.3) and 96 patients with intact ACL (40 females and 56 males, mean age ± SD 31.4 ± 7.7). The demographic data between the two groups did not show any significant difference (Table 1). The ICC and 95% confidence interval (CI) of each parameter are displayed in Table 2, all of which show excellent agreements. The body mass index (BMI), interval time from injury to surgery, TPW, and all the ratios were not conformed to normal distribution, which were depicted as median and interquartile range (IQR) in this study.

The differences in the parameters between the two groups are shown in Table 3. The TSW and TSW/TPW were smaller in the study group than in the control group (*p* < 0.001), likewise the results between male subgroups and between female subgroups (Figure 2). The LTSH/TPW was larger in the study group than in the control group (*p* = 0.015). The median value of LTSH/TPW was 12.3% (IQR 11.5–13.3%) in ACL-injured males, compared to 11.7% (IQR 10.6–12.7%) in ACL-intact males, respectively (*p* = 0.036). However, the respective difference between female subgroups was not significant (Figure 2). The LTSH/TPW (*p* = 0.007) and MTSH/TPW (*p* = 0.002) were larger in males than in females in the study group. The differences in tibial spine morphologies between males and females were not significant in the control group (*p* > 0.05) (Table 3).

The ROC curve analysis was conducted to calculate the diagnostic capacity of TSW/TPW, LTSH/TPW, and MTSH/TPW for ACL injury (Figure 3). The LTSH/TPW had a poor diagnostic ability for ACL injury (AUC < 0.7). The TSW/TPW had an AUC of 0.76 for all the patients together, 0.73 for males, and 0.80 for females, with a cutoff value of 17.0% (55.2% sensitivity and 84.4% specificity), 16.3% (60.7% sensitivity and 76.3% specificity), and 17.3% (57.5% sensitivity and 89.2% specificity), respectively.

The specific relationships between tibial spine morphologies and ACL injury were evaluated by binary logistic regression model (Figure 4). The TSW/TPW revealed significant ORs of 1.47 (95% CI [1.22–1.78], *p* < 0.001) and 1.56 (95% CI [1.23–1.98], *p* < 0.001) with regard to male and female ACL injury, respectively. In the male subgroups, LTSH/TPW was associated with ACL injury with an OR of 1.30 (95% CI [1.04–1.63], *p* = 0.020).

## 4. Discussion

The most important finding of this study was that the ratio of LTSH/TPW is a risk factor for ACL injury in males: a 1% increase in LTSH/TPW was associated with a 1.3-fold increase in the risk of ACL injury, while this relation was not significant in females. LTSH was greater in ACL-injured patients than in ACL-intact controls. In patients with ACL injury, the tibial spine height showed a significant difference between males and females.

The etiology of non-contact ACL injury is multifactorial, and the morphology of the proximal tibia is of great concern. Numerous studies have elucidated that a larger PTS and shallower medial plateau depth play pivotal roles in ACL injury [11,12,13,14,16,17]. An excessive PTS could produce a forward-shifting trend of the tibia relative to the femur, increasing the tension on the ACL [19]. Sturnick et al. [15] reported that a decreased medial tibial spine volume is associated with an increased risk of ACL injury, but such a relation was merely found in males.

Previous study has shown that males tend to have larger bony structures than females [26]. Based on our data on the TPW, individual differences, especially between males and females, were shown, which could result in unreliable results. As a result, the TPW was used to normalize these length parameters in this study. We reasoned that these ratios could reliably reflect the morphological characteristics of the tibial spine when the individual differences were significant.

In a study by Iriuchishima et al. [22], the TSW was 12.5 ± 1.9 mm in patients with ACL injury and 13.9 ± 2.1 mm in the control group, compared to 10.6 ± 1.7 mm and 12.7 ± 2.2 mm in our study, respectively. In line with the results reported by Iriuchishima et al., we revealed that the TSW was smaller in patients with ACL injury than in healthy individuals. Moreover, we also found that the ratio of TSW/TPW in the study group was significantly reduced and considered it a risk factor for ACL injury: a 1% decrease in TSW/TPW was associated with a 1.5-fold increase in the risk of ACL injury.

In accordance with the results of previous studies [14,15,22], the LTSH and MTSH did not show a significant difference between patients with and without ACL injury in this study. However, after being normalized by the TPW, the LTSH was significantly higher in patients with ACL injury than in normal subjects. Furthermore, the LTSH was demonstrated to be responsible for non-contact ACL injury in this study, indicating that a 1% increase in the LTSH/TPW was associated with a 1.3-fold increase in the risk of ACL injury. Previous research concluded that the LTSH is not implicated in ACL injury, and a potential reason was that the individual differences among participants in previous research had been neglected.

Previous studies have shown that the risk of ACL injury varies from males to females [14,15,19,27,28,29]. A decreased femoral intercondylar notch width, the height of media posterior meniscus [14], an increased PTS, and the height of the posterior portion of the lateral meniscus were merely associated with ACL injury in females [27,28]. On the other hand, Sturnick et al. [15] suggested that a decreased medial tibial spine volume was only associated with ACL injury in males and concluded that the greater variation between individuals was one of the possible reasons for such discrepancy.

In our study, significant differences in tibial spine morphologies between males and females made them at different risks for ACL injury. In patients with ACL injury, the LTSH/TPW and MTSH/TPW were greater in males than in females. Furthermore, the LTSH/TPW was greater in males with ACL injury than in males with an intact ACL. A greater LTSH/TPW was thus considered a risk factor for male ACL injury. This relationship was not statistically significant in females. On the other hand, the ratio of LTSH/TPW had a poor diagnostic capacity for ACL injury (AUC = 0.60), even in the male and female subgroups.

From the kinematic point of view, the tibial spine is closely related to the femoral intercondylar notch during knee flexion and extension [13]. Li et al. [30] applied a computer model based on double orthogonal fluorescence and magnetic resonance imaging to evaluate the kinematics of knee joints and found that during knee flexion, the contact points between the femoral condyle and tibia were located near the tibial spine. Histologically, the ACL is attached close to the intercondylar spine of the tibia [20,21]. We reasoned that an increased LTSH could contribute to an impingement of the ACL with the tibial spine, accelerating ACL injury. The impact of the increased LTSH on the impingement between the lateral tibial spine and ACL in patients with ACL injury, especially male patients, warrants further investigation.

Identifying risk factors for ACL injury reduces its incidence, promotes improvements in surgical techniques, and helps patients with individualized rehabilitation programs [14,31]. This study provides the characteristics of tibial spine morphologies by introducing the normalized height and width of the tibial spine, which are easy to obtain clinically by a simple measurement method on radiographs. In addition, the significant correlation between LTSH and male ACL injury suggests that orthopedic surgeons should pay more attention to the height of the tibial spine in males with ACL injury, especially during arthroscopy surgery.

The clinical relevance of this study lies in finding a greater LTSH in patients with ACL injuries, especially in male subjects, and demonstrating the LTSH as a risk factor for non-contact ACL injury. This, to some extent, resolves the controversy in the literature regarding the tibial spine height in patients with ACL injury and identifies the differences in tibial spine morphologies between males and females. These results also remind orthopedic surgeons of concerns during ACL reconstruction: more attention should be paid to the height of the lateral tibial spine, of male patients in particular, to avoid impingement between the grafts and lateral tibial spine. This warrants further investigation.

Our study had some limitations. First, all measurements were performed merely on radiographs, and whether data from computerized tomography images or magnetic resonance imaging would draw the same conclusion warrants further investigation. Second, the exercise intensity of all the participants included in this study is unclear. This study did not collect and analyze such data. Third, the subjects in the control group of this study were not completely healthy individuals, but they did not have ACL injury. Finally, femoral condyle morphology and other tibial anatomical parameters were not included in this study.

## 5. Conclusions

LTSH was larger in patients with ACL injury and is a risk factor for ACL injury in males. The impact of increased LTSH on the impingement between the grafts and lateral tibial spine during ACL reconstruction warrants further investigation.

## Figures and Tables

**Figure 1 ijerph-19-15589-f001:**
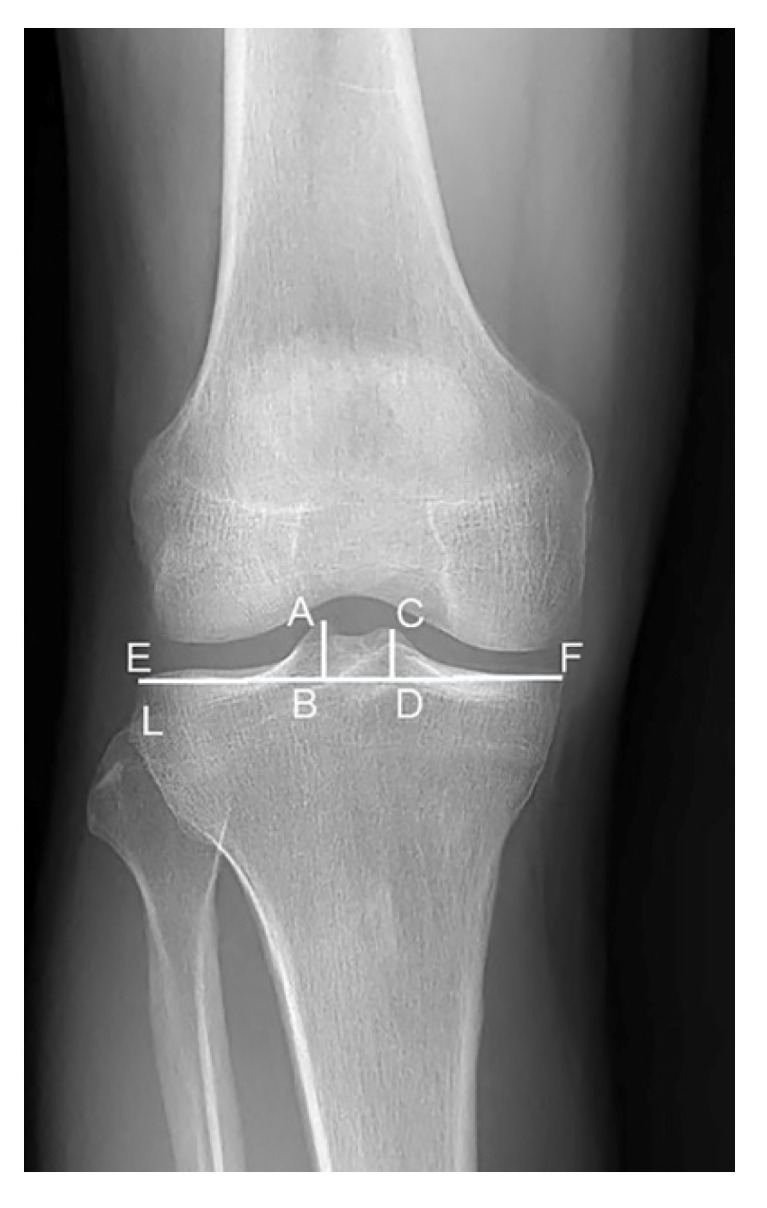
Morphological assessment of the tibial spine by standard radiograph. Tibial joint line connecting the most concave point of the medial tibial plateau and the most convex point of the lateral tibial plateau is shown (Line L). The length between the intersections of the L-line with the tibia condyles (E and F) is regarded as the tibial plateau width (TPW). The distances between the top of the tibial spine (A and C) and line L is regarded as the lateral tibial spine height (LTSH) or medial tibial spine height (MTSH). BD is considered as the tibial spine width (TSW).

**Figure 2 ijerph-19-15589-f002:**
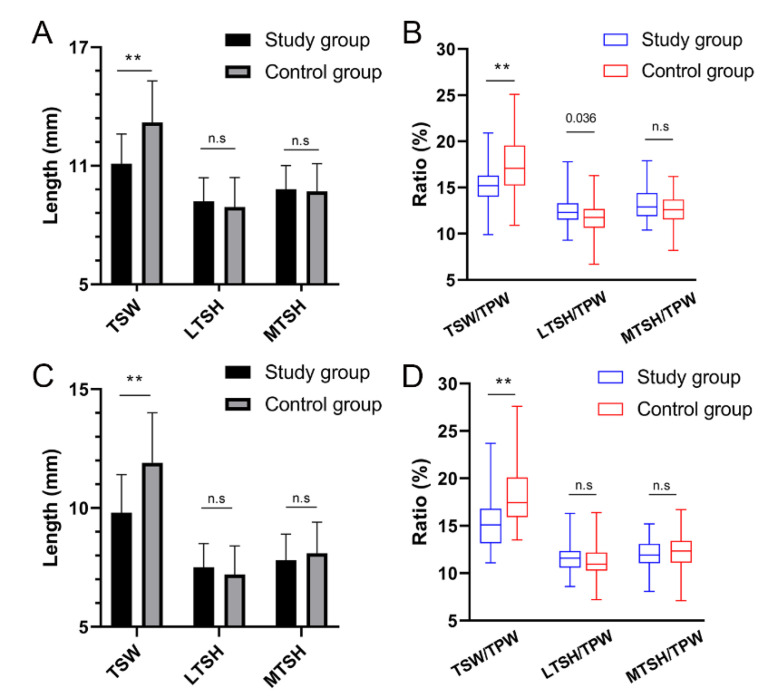
Differences in tibial spine morphologies between male and female subgroups. (**A**,**B**) show the results in males between the study group and control group. (**C**,**D**) show the results in females between the study group and control group. TSW, tibial spine width; LTSH, lateral tibial spine height; MTSH, medial tibial spine height; TPW, tibial plateau width; **, statistically significant with a *p* value of <0.001; n.s., not statistically significant.

**Figure 3 ijerph-19-15589-f003:**
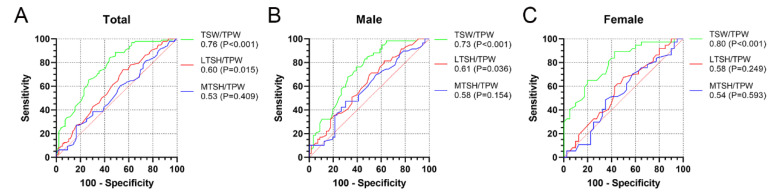
Receiver operating characteristic curves of three normalized tibial spine width or height in all the patients together (**A**), in males (**B**), and in females (**C**). The area under the curve (*p* value) is shown, with a value greater than 0.7 indicating fair to good diagnostic capacity. TSW, tibial spine width; LTSH, lateral tibial spine height; MTSH, medial tibial spine height; TPW, tibial plateau width.

**Figure 4 ijerph-19-15589-f004:**
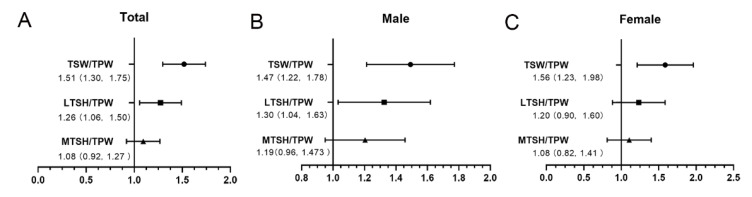
The specific relationships between tibial spine morphologies and ACL injury in all the patients together (**A**), in males (**B**), and in females (**C**), showing the odds ratio and a 95% confidence interval. TSW, tibial spine width; LTSH, lateral tibial spine height; MTSH, medial tibial spine height; TPW, tibial plateau width.

**Table 1 ijerph-19-15589-t001:** Demographic data of the included subjects.

	Study Group (*n =* 96)	Control Group (*n =* 96)	*p* Value
Age, mean ± SD, years	29.5 ± 8.3	31.4 ± 7.9	0.093
Sex, Male/Female, *n*	59/37	56/40	0.659 ^a^
Side, Left/Right, *n*	50/46	46/50	0.564 ^a^
BMI			0.901 ^b^
Median	23.44	23.49	
IQR	4.86 (21.22–26.08)	4.11 (21.46–25.57)	
Interval Time, days			0.361 ^b^
Median	16.50	16.50	
IQR	24.75 (7.25–32.00)	21.00 (10.00–31.00)	

SD, standard deviation; BMI, body mass index; IQR, interquartile range; Interval Time, the time from injury to surgical treatment; ^a^, the results of the chi-square test; ^b^, the results of Wilcoxon rank-sum test.

**Table 2 ijerph-19-15589-t002:** Inter-observer reliability analysis and mean ± SD of each anatomic parameter.

Variables	Study Group			Control Group		
	Observer 1	Observer 2	ICC (95% CI)	Observer 1	Observer 2	ICC (95% CI)
Tibial Spine Height						
Lateral, mm	8.4 ± 1.4	8.6 ± 1.5	0.873 (0.815, 0.913)	8.1 ± 1.6	8.3 ± 1.7	0.922 (0.885, 0.947)
Medial, mm	9.0 ± 1.5	9.1 ± 1.5	0.901 (0.856, 0.933)	9.0 ± 1.6	9.1 ± 1.7	0.930 (0.897, 0.953)
Tibial Plateau Width, mm	70.2 ± 5.8	69.9 ± 5.6	0.930 (0.897, 0.953)	71.5 ± 6.7	71.9 ± 6.8	0.960 (0.940, 0.973)
Tibial Spine Width, mm	10.6 ± 1.8	10.6 ± 1.6	0.891 (0.841, 0.926)	12.7 ± 2.3	12.6 ± 2.2	0.899 (0.853, 0.932)

SD, standard deviation; CI, confidence interval; ICC, intraclass correlation coefficient, with a value more than 0.75 indicating excellent agreement.

**Table 3 ijerph-19-15589-t003:** Comparison of the variables between subgroups, showing the mean ± SD or median (IQR).

Variables	Study Group (*n =* 96)				Control Group (*n =* 96)				*p* Value ^a^
	Male (*n =* 59)	Female (*n =* 37)	*p* Value	Total	Male (*n =* 56)	Female (*n =* 40)	*p* Value	Total	
LTSH, mm	9.2 ± 1.2	7.5 ± 1.0	**<0.001**	8.5 ± 1.4	8.9 ± 1.5	7.2 ± 1.2	**<0.001**	8.2 ± 1.6	0.172
MTSH, mm	9.8 ± 1.2	7.8 ± 1.1	**<0.001**	9.0 ± 1.5	9.7 ± 1.4	8.1 ± 1.3	**<0.001**	9.1 ± 1.6	0.929
TSW, mm	11.1 ± 1.5	9.8 ± 1.6	**<0.001**	10.6 ± 1.7	13.2 ± 2.1	11.9 ± 2.1	**0.003**	12.7 ± 2.2	**<0.001**
TPW, mm			**<0.001**				**<0.001**		0.055 ^b^
Median	73.8	63.9		71.1	76.1	65.3		73.2	
IQR	5.5 (71.3–76.8)	5.0 (61.9–66.9)		9.5 (65.2–74.7)	4.3 (74.2–78.5)	4.7 (62.9–67.6)		10.0 (66.5–76.5)	
TSW/TPW, %			0.839				0.220		**<0.001** ^ **b** ^
Median	15.2	15.1		15.2	17.1	17.4		17.3	
IQR	2.3 (14.0–16.3)	3.6 (13.1–16.7)		3.0 (13.5–16.5)	4.3 (15.2–19.5)	4.2 (15.9–20.1)		4.0 (15.6–19.6)	
LTSH/TPW, %			**0.007**						**0.015** ^ **b** ^
Median	12.3	11.6		12.0	11.7	11.0	0.081	11.6	
IQR	1.8 (11.5–13.3)	1.7 (10.6–12.3)		1.9 (11.0–12.9)	2.1 (10.6–12.7)	1.9 (10.3–12.2)		2.0 (10.4–12.4)	
MTSH/TPW, %			**0.002**						0.409 ^b^
Median	12.9	11.9		12.6	12.6	12.4	0.336	12.5	
IQR	2.5 (11.9–14.4)	2.0 (11.0–13.0)		2.5 (11.6–14.1)	2.2 (11.5–13.7)	2.3 (11.1–13.4)		2.2 (11.4–13.6)	

LTSH and MTSH, lateral and medial tibial spine height, respectively; TSW, tibial spine width; TPW, tibial plateau width; SD, standard deviation; IQR, interquartile range. ^a^, the results of Wilcoxon rank-sum test; ^b^, the *p* value of the difference in the total value of each anatomical parameter between the study group and the control group; bold indicates statistical significance.

## Data Availability

The data associated with the paper are not publicly available but are available from the corresponding author upon reasonable request.

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
