# Peer review of "Tibial Spine Height Measured by Radiograph Is a Risk Factor for Non-Contact Anterior Cruciate Ligament Injury in Males: A Retrospective Case-Control Study"

_ijerph, 2022, doi:10.3390/ijerph192315589_

Round 1
Reviewer 1 Report
There are a few grammatical errors. Please check this. Statement that males have higher exertion level than female seems incorrect and offensive. There are similar exertion levels but as you demonstrated in your study there are anatomic differences between males and females. I would remove this statement. At the beginning of the discussion it is stated that the differences in LTSH is the most significant finding but I would submit the that the OR for ACL rupture as demonstrated by the LTSH/TSW % is the most significant finding. The study is sound and the findings are significant and warrant further study particularly in relation to ACL reconstruction techniques and risk of failure of ACLr.
Reviewer 2 Report
The authors perform a simple job. They first compare data obtained by two orthopedic surgeons. Then they compare two groups of people to see differences between them.
The authors are aware of the limitations of their work and show this in the discussion. Despite this, the methodology is correct and the interpretation of the results as well.
In the method section, a subsection indicating the design of the study and/or the experimental protocol is missing. Before the radiographic technique. It is confused.
Line 62. I don't see the description of the acronym MRI.
I don't understand the usefulness of work for injury prevention.
